# Challenges and Suggestions in the Management of Stomach and Colorectal Cancer in Uzbekistan: The Third Report of the Uzbekistan–Korea Oncology Consortium

**DOI:** 10.3390/ijerph20085477

**Published:** 2023-04-12

**Authors:** Chai Hong Rim, Won Jae Lee, Odiljon Akhmedov, Ulugbek Sabirov, Yakov Ten, Yakhyo Ziyayev, Mirzagaleb Tillyashaykhov, Jae Suk Rim

**Affiliations:** 1Department of Radiation Oncology, Korea University Ansan Hospital, Korea University, Seoul 15355, Republic of Korea; 2Department of Healthcare Management, Gachon University, Seongnam-si 13120, Republic of Korea; 3Department of science, Republican Specialized Scientific Practical-Medical Center of Oncology and Radiology, Farobiy Street 383, Tashkent 100179, Uzbekistan; 4Ministry of Health of the Republic of Uzbekistan, Tashkent 100011, Uzbekistan; 5Department of Oral and Maxillofacial Surgery, Guro Hospital, Korea University College of Medicine, 148 Gurodong-ro, Guro 2-dong, Guro-gu, Seoul 08308, Republic of Korea

**Keywords:** gastric cancer, stomach cancer, colorectal cancer, Uzbekistan, South Korea

## Abstract

In general, as the national standard of living and life expectancy of people increase, the health burden of cancer also increases. Prevention strategies, including the screening and investigation of the causes of cancer as well as the expansion of treatment infrastructure, are necessary. In this review, we discussed the management strategies for gastric and colorectal cancers in Uzbekistan. Gastrointestinal cancers can be significantly prevented by certain screening strategies such as endoscopic examination. Furthermore, as both cancer types are closely related to the eating habits and lifestyles of people in Uzbekistan, such causes should be investigated and prevented. Practical advice to increase the efficiency of treatment is included, considering the current situation in Uzbekistan. Data from South Korea, which has performed nationwide screening for two decades and has made progress in improving the prognosis of patients with gastrointestinal cancers, will be discussed as a literature control.

## 1. Introduction

In general, as society develops, the health burden of cancer increases [1]. The incidence of cancer increases significantly after the age of 65 years and peaks in the 8th and 9th decades of life [2]. Developed society has a longer life expectancy and more attempts to diagnose cancer, which increases the cancer incidence. In recent decades, Uzbekistan has shown socioeconomic development [3], and the health burden of cancer in Uzbekistan is expected to increase accordingly. The age-standardized cancer incidence per 100,000 people in Uzbekistan is 108.1, which is relatively low globally [4,5]. However, the mortality per incidence rate is as high as 67%, which is similar to the figures of sub-Saharan African regions (69.2–70.6%). The corresponding incidence and mortality per incidence rates in South Korea were 242.7 and 31.1% (Figure 1).

Gastric cancer (GC) and colorectal cancer (CRC) are serious diseases that rank as the first and fifth causes of cancer mortality in Uzbekistan, respectively. The risk of gastric cancer (GC) or colorectal cancer (CRC) is closely related to lifestyle and diet [6]; therefore, social efforts are necessary to identify the causes and reduce cancer risks. However, research investigating the cause of cancers in Uzbekistan is lacking. Therefore, we will discuss health measures based on an international literature review, anticipating the cause of these cancers in Uzbekistan. GC and CRC are diseases that can be significantly benefited by preventive screening, such as endoscopy or fecal tests [7,8,9]. Since nationwide screening has not been performed in Uzbekistan, we will review the benefits of screening in major countries and discuss the necessity of cancer screening in Uzbekistan. Suggestions to increase the efficiency of the treatment of GC and CRC will also be discussed in consideration of social and economic situations. Data from South Korea, which conducted a nationwide screening for these cancer types over the past decades and developed an internationally standardized treatment infrastructure, were analyzed as a main comparative control. The target audiences of our analytical document are policymakers and healthcare providers, as well as oncologists. The purpose of this review is to provide references for prioritizing health policy in Uzbekistan, to suggest the direction of domestic research, and to summarize and present comprehensive knowledge about prevention, screening, and treatment for healthcare providers. 

**Figure 1 ijerph-20-05477-f001:**
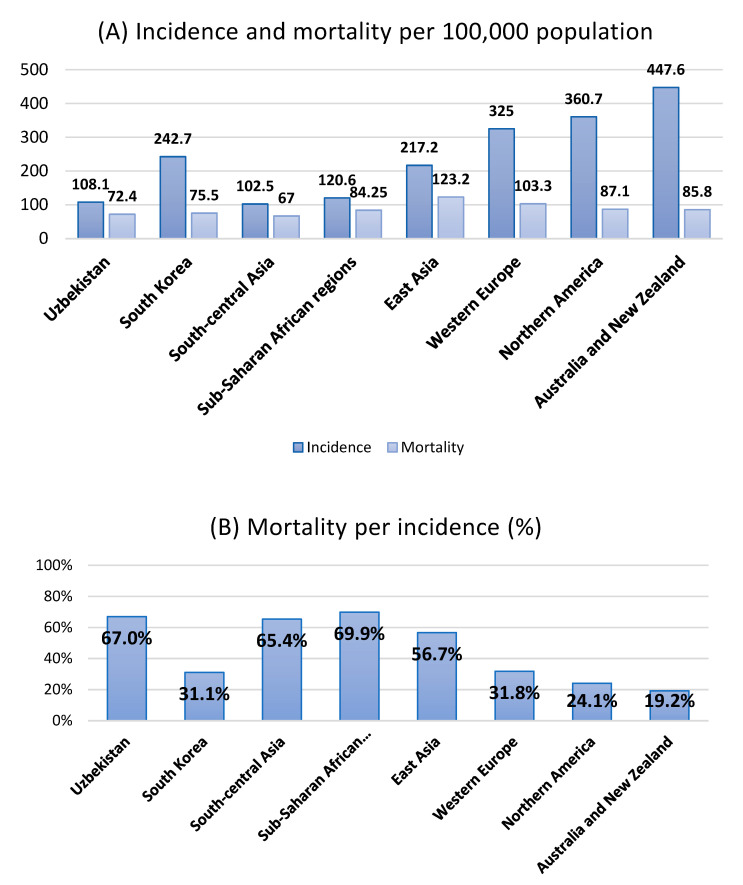
(**A**) The global incidence and mortality of all cancers per 100,000 population according to regions. (**B**) Mortality per incidence (age-standardized mortality per 100,000 people/corresponding incidence) percentile according to regions. Data source: Data on the incidence and mortality are obtained from the 2020 GLOBOCAN [4,5,10].

## 2. Gastric Cancer

### 2.1. Epidemiology

GC is the fifth most common cancer type and the fourth leading cause of cancer-related deaths worldwide [5]. The prevalence of GC is high in Asia. The highest prevalence is in East Asia and Southeast Asia and is the leading cause of mortality in Central Asian regions such as Iran, Afghanistan, Kyrgyzstan, and Uzbekistan [5]. Its incidence in men is approximately twofold higher than that in women [5]. The incidence of GC in Uzbekistan tis 9.8 per 100,000 people (age-standardized rate), and the corresponding mortality rate is 7.9. The age-standardized incidence rate of GC in South Korea is as high as 27.9 per 100,000 people. However, the corresponding mortality rate was relatively low at 6.1. When calculating the incidence per mortality rate (age-standardized mortality per 100,000 people/corresponding incidence), the figures for Uzbekistan and South Korea were 80.6% and 21.9%, respectively, indicating a significant difference [4,5,10]. In South Korea, gastroscopy or gastrography has been performed every 2 years since 1999 for all citizens aged 40 years and older [11]. In South Korea’s National Cancer Screening Report, national screening using gastroscopy reduced the GC mortality rate by 57% [8]. We assume that the difference in incidence per mortality is mainly due to the effect of screening. By region, the incidence and mortality rates per 100,000 people in Central Asia, to which Uzbekistan belongs, were 5.5 and 4.8, respectively. In sub-Saharan Africa, the corresponding rates were 4.1–4.5 and 3.7–4.0, respectively, and the incidence per mortality was close to 90%. In East Asia, Western Europe, North America, Australia, and New Zealand, the incidence rates per 100,000 people were 22.4, 5.9, 4.2, and 4.5, respectively, while the mortality rates were 14.6, 3.3, 1.8, and 2.1, respectively. The incidence per mortality was 46–65%. Table 1 summarizes the global epidemiological data [4,5,10].

The prevalence of GC in Uzbekistan was 10.8%, and it is the second most common cancer type and the leading cause of death (12.8%) (Figure 2). From an international perspective, the incidence of GC in Uzbekistan appears low. However, considering the higher mortality and prevalence compared with those of other cancer types in Uzbekistan, the incidence can increase in the future with the expansion of cancer screening and the aging of the population. Therefore, a prevention strategy should be established in advance by exploring the possible risk factors for GC in the Uzbek population.

### 2.2. Risk Factors and Diet

The factors related to the onset of GC include *Helicobacter pylori* infection and gastritis, a high-salt diet, obesity, smoking, and drinking alcohol. According to a meta-analysis study by Yang et al. [14] including 9500 GC patients, the incidence of GC was 22% higher in obese populations with a body mass index (BMI) of ≥25 kg/m^2^ than in those with a normal BMI. A recent meta-analysis reported that the incidence of GC in the high-salt diet group increased by 20–25% [15]. In a large study involving 10 European countries, smokers had a 45% increased risk of developing GC compared with nonsmokers, and the risk of GC became similar to that of nonsmokers 10 years after quitting smoking [16]. Individuals who drank four or more drinks of alcohol per day had a 20% higher risk of developing stomach cancer compared with that of non-drinkers [17]. The incidence of GC among moderate drinkers who drink two to three glasses or less per day remains controversial [18]. Some studies reported that the increased risk of gastric cancer varies depending on the type of alcohol, but the results are inconsistent in the literature [19,20,21,22]. The World Cancer Research Fund and the American Institute of Cancer Research’s third expert report stated that the risk of cancer increases with the amount of ethanol, regardless of the type of alcohol [6]. Smoking cessation is necessary to prevent stomach cancer, and the general public should be advised that the increased consumption of high-salt foods can cause cancer.

Stomach cancer is associated with its pathogenesis as well as food intake. Therefore, the eating habits and risk factors for GC of people in Uzbekistan should be investigated. High salt intake increases *H. pylori* colonization by causing inflammation and atrophy of the stomach wall [23]. In addition, fermented vegetables (consumed mainly in Asia) cause the endogenous formation of N-nitrosamine, a carcinogen [24]. According to a recent report by the World Cancer Research Fund and the American Institute for Cancer Research, every 0.5 serving (20 g) per day of salted vegetables increased the risk of GC by approximately 9% [6]. A previous meta-analysis reported that the population group that frequently consumed salted fish had a 24% increase in the incidence of GC [25]. Consumption of processed meat (e.g., ham, sausage, minced meat, and bacon) can lead to the formation of carcinogenic N-nitroso compounds in the stomach. Heme iron found in meat can also stimulate the endogenous formation of N-nitroso compounds and cause cancer [26]. Consumption of grilled or barbecued meat induces the formation of known carcinogens, such as polycyclic aromatic hydrocarbons and heterocyclic amines [24]. In a Japanese study, the population group that frequently consumed grilled meat had a 2.27-fold higher risk of GC mortality than the control group [27]. By contrast, the consumption of fruits reduces the incidence of GC by reducing inflammation and DNA damage caused by *H. pylori* infection [6]. According to a previous dose–response meta-analysis by Wang et al. [28], the incidence of GC decreased by 5% for every 100 g of fruit consumed.

As mentioned above, the onset of GC is closely related to diet, in addition to the known clinical factors such as gastritis caused by *H. pylori* infection, smoking, drinking, and obesity. The Uzbekistan people’s diet tends to be meat-heavy, and they often consume grilled or fried meat, such as plov and shashlik. However, the Uzbekistan people’s diet also includes a variety of vegetables and fruits [29]. Dietary guidance is needed to reduce the intake of grilled or processed meat and increase the intake of fruits and vegetables.

### 2.3. Helicobacter Pylori and the Need for Early Screening

*H. pylori* resides in the stomach wall. It was designated as a class 1 carcinogen in 1994 by the International Agency for Research on Cancer (IARC). According to a large-scale study conducted in thirteen countries, the incidence of GC was approximately six times higher in the *H. pylori*-infected population [30]. In a Japanese study, after 7–8 years of follow-up of 1526 patients with gastritis or duodenitis (1246 *H. pylori* infected and 280 non-infected), 36 (2.9%) developed GC, all of whom were *H. pylori* infected. The prevalence of *H. pylori* varies widely worldwide; however, in general, the prevalence is lower in developed and Western countries (11% in Northern Europe, 30% in the US, 50% in Korea, and 72%–82% in South America). This is because the *H. pylori* bacteria is estimated to be transmitted through a close group or person-to-person contact within the family [31]. No academic data on the prevalence of *H. pylori* in Uzbekistan could be found; in a study of a small population in Kazakhstan [32], 80% of the regional population was infected with *H. pylori*, and the prevalence was related to drinking river water (not tap or well water) or low socioeconomic status. The prevalence of *H. pylori* could be high in Uzbekistan, considering that Uzbekistan and Kazakhstan are neighboring countries and have similar cultures with frequent communications. 

*H. pylori* causes atrophic gastritis and progressively causes cancer through repeated metaplasia [33]. According to Correa’s cascade of gastric carcinogenesis, H. pylori infection and salt intake could cause chronic gastritis, which then progresses to carcinoma through intestinal metaplasia and dysplasia due to increased intragastric pH, anaerobic bacterial growth, and inflammatory response [34,35]. Gastroscopy can detect malignancies and precancerous lesions that have not yet become cancerous but are likely to develop into cancer. In South Korea, gastroscopy or gastrography has been performed every 2 years on all citizens aged 40–74 since 1999. According to a study conducted by Cho et al. [8]., gastroscopy in Korea reduces the GC mortality rate by 57%. In prospective Japanese studies, early screening through gastroscopy reduced GC mortality rate by 65% [36,37]. Screening using gastrography reduced GC mortality rate by 45% in the combined analysis of Japanese studies, but no significant decrease in mortality was observed in a previous Korean study (Cho et al. [8]) [11]. 

Regarding the period of screening, the tendency for the incidence to increase after the age of 40 should be considered [11,38]. In a Korean nationwide study [8], a decrease in mortality following GC screening was observed from the age of 40–74 (odds ratio: 0.60–0.85). However, a significant decrease in mortality was not observed in the age of 75–84, whereas an increase in mortality rate was observed in those aged ≥85 years (odds ratio: 2.15). In a Japanese study, the relative risk of death in the population who underwent screening at the age of 40–59, was significantly reduced (risk ratio: 0.3–0.6) [39]. Considering the average life expectancy of Uzbekistan (72 years) [40], screening from the age of 40 could be recommended and discussion among domestic experts is required for the age of termination.

Therefore, early screening through gastroscopy (gastrography can be performed considering the required manpower and its efficacy) is an essential policy in Uzbekistan to detect GC early and to increase the survival rate.

### 2.4. Treatment Consideration

The primary treatment for GC is surgical resection. In Korea, GC screening has been widely implemented, and the detection rate of early disease has relatively increased. Lee et al. [41] reported that the prevalence of early-stage GC increased from 54% (2003–2007) to 63.5% (2008–2012) and 81% (2013–2018) in a national university hospital. As cancer screening continues to expand in Uzbekistan, the role of endoscopic mucosal resection is becoming important. In most institutions, endoscopes are washed with disinfectant solution and water, and automatic sterilizers are essential to prevent the spread of H. pylori and hepatitis virus [42].

Almost all national cancer centers in Uzbekistan can provide surgical treatment, but only a few of the main centers can provide chemotherapy or radiation therapy [42]. Perioperative chemotherapy is generally recommended for patients with T2 stage or higher or lymph node metastasis [43]. Conventional chemotherapy, using fluorouracil or platinum-based drugs, are mainly used for GC treatment [44]; therefore, the associated economic burden is relatively small. Tillyashaykhov et al. [45] reported that the availability of first-line drugs for all cancer types significantly increased from 30.3% in 2016 to 95.1% in 2018. The addition of trastuzumab increased the effectiveness of the patients’ treatments, but the survival rate only demonstrated a moderate increase (median survival 13.8 vs. 11.1 months); this could be economically burdensome in developing countries [46,47].

Radiation therapy is used for patients in whom complete resection has not been achieved or for those with advanced disease stage as a palliative treatment [43]. The effects of chemoradiation and those of chemotherapy were comparatively analyzed after D2 resection, which is the current standard surgery, but the additional benefit of radiotherapy was not shown [48]. However, because the prevalence of advanced GC remains high in Uzbekistan, radiation therapy can be applied to patients who have undergone suboptimal resection or as a palliative modality for symptom relief.

### 2.5. Summary and Suggestions

GC is one of the causes of cancer-related deaths in Uzbekistan. Although the international incidence is only moderate, the fatality rate is relatively high. This finding suggests that most diseases are diagnosed at an advanced stage, indicating a dire need for screening. In South Korea, national GC screening has reduced the GC mortality rate by 57%. Hence, nationwide screening using gastroscopy (or gastrography) is urgently needed in Uzbekistan. Moreover, automatic endoscope sterilizers should be provided as soon as possible. The causes of GC in Uzbekistan, such as *H. pylori* infection, consumption of a high-salt diet, obesity, and excessive meat consumption, should be investigated to establish effective preventive strategies. Since the mainstay of treatment is surgery, specialized surgeons must be trained, and a sufficient surgical capacity must be ensured. The availability of first-line systemic treatment has improved significantly in Uzbekistan. As the proportion of advanced GC remains high in Uzbekistan, radiation therapy can be applied to patients who have undergone suboptimal resection or as a palliative modality for symptom relief.

## 3. Colorectal Cancer

### 3.1. Epidemiology

In Uzbekistan, CRC ranked 4th (6.7%) in terms of prevalence and 5th in terms of cancer-related mortality rate (6.1%) (Figure 2) [12]. CRC recently ranked 1st or 2nd in terms of prevalence and 2nd or 3rd in terms of cancer-related mortality rate in Korea [49]. From an international perspective, Central Asia and Africa have the lowest proportions of colorectal cases. In general, the prevalence of CRC and the human development index tend to be proportional. This is because the prevalence of CRC is directly related to changes in lifestyle—for example, meat consumption, sedentary lifestyle, and obesity—that occur as the standard of living increases [5]. Therefore, the health burden of CRC in Uzbekistan has gradually increased with the recent economic development. The age-standardized incidence per 100,000 people in Uzbekistan is 8.9, which is relatively low compared with the international incidence rate. The mortality rate was 5.2, which corresponded to the calculated incidence per mortality of 58.4%. In South Korea, the age-standardized incidence per 100,000 people is 27.2, and the corresponding mortality is 7.8; therefore, the incidence per mortality is calculated to be 28.7%. In sub-Saharan Africa, the incidence per 100,000 people is 6.7–7.9, and the corresponding mortality is 5.1–5.4; therefore, the incidence per mortality is approximately 68%–76%. In Western Europe, North America, Australia, and New Zealand, the incidence per 100,000 people is as high as 26–33, while the incidence per mortality is approximately 28.6–35.5%. The international incidence data are summarized in Table 2. 

### 3.2. Risk Factors and Lifestyle

CRC is closely associated with lifestyle and related diseases. Several meta-analyses have summarized a number of studies investigating the association between lifestyle and CRC risks. Obesity is an important cause of CRC. The incidence of CRC increased by 19% in obese individuals (BMI ≥ 30 kg/m^2^) compared with that in normal-weight people (BMI < 25 kg/m^2^). In a dose–response relationship analysis, a 2 kg/m^2^ increase in BMI was associated with a 7% increase in CRC risk [50]. Similarly, physical activity reduces the risk of CRC by approximately 24% [51]. Smokers had an 18% higher risk of CRC and a 25% higher mortality rate from CRC compared with that of nonsmokers [52]. With regard to alcohol intake, the incidence of CRC among heavy drinkers (4 or more drinks per day) was approximately 52% higher, while those who consumed 2–3 drinks had a 21% higher incidence of CRC; those who drank one cup or less per day had no significant increase in CRC risk [52]. Diabetes has also been strongly associated with CRC. Patients with diabetes mellitus had a 38% pooled increase in the risk of developing colon cancer and a 20% increased risk of developing rectal cancer [53].

CRC is also closely associated with dietary intake. The third expert report from the World Cancer Research Fund and American Institute of Cancer Research [6] comprehensively reviewed the available literature, including a dose–response relationship analysis. The consumption of red meat was associated with an increased risk of CRC; the risk was estimated to increase by 12% for every 100 g/day increase in intake. Consumption of processed meat such as ham or sausage is also correlated with the risk of developing CRC. It was estimated to increase by 16% for every 50 g/day increase in intake. On the contrary, the consumption of whole grains significantly reduced the risk of CRC by 17% for every 90 g/day intake. The risk of CRC was inversely correlated with fish intake. The risk decreased by 11% per 100 g/day increase in dietary intake. Consumption of dairy products decreased the risk of CRC, and every 400 g/day increase in the intake of dairy products was associated with a 13% decrease in CRC.

As such, CRC is related to various lifestyles and eating habits such as drinking, smoking, diabetes, meat intake (negative factors), exercise, whole grain intake, dairy intake, and fish intake (positive factors). Studies regarding the causes of CRC in Uzbekistan are currently lacking. Hence, the investigation of all risk factors associated with CRC in Uzbekistan is necessary to establish guidelines on the appropriate lifestyle and diet for CRC prevention.

### 3.3. Necessity of Screening

Screening using colonoscopy can detect CRC early and reduce mortality rate. In a population screened with colonoscopy, the mortality rate due to CRC was reduced by 65–88% [7,54,55]. However, the effectiveness of colonoscopy is affected by the operator’s skill, and serious complications can occur in a few cases. Colonoscopy may cause serious side effects such as perforation or bleeding, generally in 0.5% of patients who underwent this procedure [56,57]. In the European Union’s quality assurance report, the incidence of serious complications ranged from 0% to 0.3% in well-organized and high-quality institutions [58]. Bressler et al. [59] reported that higher proportions of new or missed CRCs after colonoscopy were found on examinations performed by an internist or family physician (~1.8 times higher than that performed by a gastroenterologist).

Previous landmark studies conducted in Western countries reported that the occult blood test reduced CRC mortality rate by 15–33% [9]. Although fecal occult blood testing has less clinical benefit in magnitude compared with that of colonoscopy, it is a simple screening method and does not cause serious complications. By contrast, the fecal occult blood test has a high false-positive and false-negative rate, which may require retesting or cause psychological stress to the examinee [9]. Although the oncologic benefit might be smaller, occult blood testing could be useful for nationwide screening owing to its higher compliance compared with that of colonoscopy. In a randomized study (colonoscopy vs. 3 occult blood tests in 1 year) conducted in the United States, 67% of the occult blood test population underwent the recommended screening, but only 38% of the colonoscopy population completed the task [60]. Therefore, considering the degree of compliance and effectiveness of screening, the utility of fecal occult blood testing is not necessarily lower than that of colonoscopy as a nationwide screening modality [61]. 

In Uzbekistan, the number of colonoscopy procedures remains low, and most video systems are not equipped with endoscopes. In addition, very few personnel are professionally trained in performing endoscopy. As water and antibacterial soap are only used to clean the endoscope, an automatic sterilizer should be provided for sterilization, and the endoscope equipment should be developed further [42]. Expanding the colonoscopy equipment and increasing the number of specialists can yield significant clinical benefits in Uzbekistan. The benefits of nationwide screening using fecal occult blood testing should also be discussed.

The incidence of CRC increases after the age of 40, with 90% occurring after 50 [62]. Therefore, CRC screening is recommended to begin at the age of 45–50 in major countries. Choi et al. [63]. reported that starting screening at the age of 45, rather than 50, increased the sensitivity of CRC detection by 5%. In South Korea, annual occult blood testing is recommended as a national screening for people aged of 45–80 [9]. The USPSTF (United States Preventive Services Task Force) recently updated the starting age of screening from 50 to 45 years, and recommends CRC screening (blood test or colonoscopy) for people aged of 45–75 years, and that screening should be performed selectively for those aged of 76–85 years [64,65]. EU guidelines recommend fecal immunochemical test screening for those aged 50–74 [66]. Considering the average life expectancy of Uzbekistan (72 years) [40], starting screening around the age or 45 could be recommended, and discussion of domestic experts is required for the age of termination.

### 3.4. Treatment Suggestion

Although surgery is the mainstay of treatment for CRC, a recent IARC imPACT report suggested the necessity of multidisciplinary treatment discussions, including preoperative radiation therapy [42]. In landmark randomized studies, preoperative radiotherapy significantly lowered the recurrence rate from one-half to one-third [67,68]. Radiation therapy is effective for tumors located close to the anal margin and can improve the quality of life of patients by increasing the sphincter preservation rate [69]. Conventional drugs such as 5-fluorouracil based regimens are mainly used for chemotherapy; therefore, their cost burden is less significant. Tillyashaykhov et al. [45]. reported that at the 2019 ESMO Congress, the availability of first-line drugs for all cancers significantly increased from 30.3% in 2016 to 95.1% in 2018. Targeted anticancer drugs, such as bevacizumab, cetuximab, and panitumumab, have shown effectiveness in several studies, but some studies failed to achieve the target endpoints or reported increased toxicities [70]. Considering that the cost burden of targeted anticancer drugs is significantly higher than that of conventional drugs, this treatment might be difficult to implement among cancer patients in Uzbekistan.

### 3.5. Summary and Suggestions

In Uzbekistan, the prevalence of CRC was 6.7%, while the mortality rate from CRC was 6.1%. Although its global incidence is currently low, as the incidence of CRC increases as the country develops, the frequency will likely increase in Uzbekistan as well. Nationwide screening using fecal occult blood testing is recommended, as it can increase the early detection of CRC with high applicability and without possible complications, and it can reduce the mortality rate. Screening through colonoscopy can be more efficient, but the equipment and manpower required need to be expanded further. Radiation therapy can be considered for the treatment of rectal cancer to lower the recurrence rate and increase the sphincter preservation rate. Among the diets and lifestyles of the Uzbekistan people, those related to the development of CRC should be investigated and discussed to avoid them.

## 4. General Summary and Recommendations

GC is the leading cause of cancer-related deaths in Uzbekistan (12.9%) [12]. CRC ranks fifth among the leading causes of cancer death, but its prevalence is expected to increase as the standard of living increases [5]. Stomach and CRCs have significant benefits from early screening, especially when endoscopes are used. In addition, since these cancer types are closely related to the diet and lifestyle, their causes should be domestically investigated and prevented. For both cancer types, surgery is the mainstay of treatment; for rectal cancer, adjuvant radiotherapy can help reduce recurrence. Although several novel targeted agents have been reported to be effective, classical cytotoxic chemotherapy has been the mainstay of systemic treatment. 

We suggest the following recommendations based on the above literature analysis. The suggestion is classified as a strong recommendation if the impact is significant and the possibility of implementation is high. The suggestion is classified as a recommendation if it has a significant impact but has a moderate feasibility. Even if the impact of the suggestion is significant, it is classified as a recommendation if it requires a mid- to long-term plan or if it is rather an academic proposal.

Early screening using gastroscopy reduced GC mortality by 57% in a Korean study and 65% in a Japanese study. The introduction of national screening using gastroscopy in Uzbekistan is necessary (Strongly recommended). Stomach cancer is related to lifestyle habits such as obesity, smoking, excessive drinking, and a high-salt diet; education to prevent them is necessary (Recommended). Since the prevalence of H. pylori, the most important cause of GC, is also high in Uzbekistan, the necessity and social feasibility of antibacterial therapy should be discussed (Recommended).

Nationwide screening using colonoscopy is recommended considering its significant effect, which reduces the cancer mortality rate of 65–88%. Given the possible rare but serious complications, standardized colonoscopy screening systems and the increased training of specialized physicians are required (Strongly recommended). Since colorectal cancer is related to lifestyle habits such as meat eating, a sedentary lifestyle, and obesity, education to prevent them is necessary (Recommended). Increasing the application of preoperative external radiotherapy to reduce the recurrence rate of colorectal cancer is suggested (Recommended).

The limitations of this review are as follows: This review aims primarily to summarize comprehensive knowledge and advice on prioritizing health policy. However, this review may not be detailed enough for clinicians to refer to for actual treatment or diagnosis. Furthermore, we suggested the priority health needs in consideration of the socio-economic situation of Uzbekistan. Therefore, topics related to the latest treatment or diagnosis that require a significant economic burden may not have been addressed enough. Although data are currently lacking, practice guidelines should be based on the experience of clinicians in Uzbekistan and domestic research data. Further details should be addressed in specific practice guidelines by professional groups including domestic clinicians. 

## Figures and Tables

**Figure 2 ijerph-20-05477-f002:**
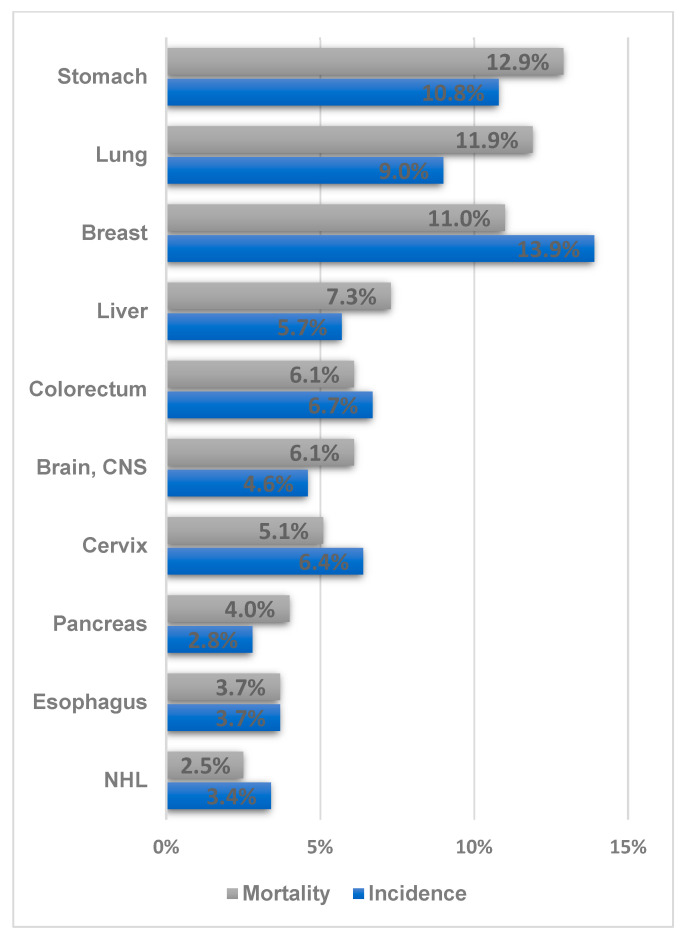
Ranking of the frequency and mortality of the major cancer types in Uzbekistan (Data source: Cancer country profile, WHO, 2020; Global Health Observatory, WHO, 2016. Figure drawn by authors) [12,13].

**Table 1 ijerph-20-05477-t001:** Brief global statistics of stomach cancer, including Uzbekistan and South Korea.

	Uzbekistan	South Korea	South-Central Asia	Sub-Saharan African Regions	East Asia	Western Europe	Northern America	Australia and New Zealand
Incidence	9.8	27.9	5.5	4.1–4.5	22.4	5.9	4.2	4.5
Mortality	7.9	6.1	4.8	3.7–4.0	14.6	3.3	1.8	2.1
Mortality per incidence	80.6%	21.9%	87.3%	88.9–90.2%	65.2%	55.9%	42.9%	46.7%

All values indicate the age-standardized rates per 100,000 population. Data source: Data on the incidence and mortality are obtained from the 2020 GLOBOCAN [4,5,10].

**Table 2 ijerph-20-05477-t002:** Brief global statistics of colorectal cancer, including Uzbekistan and South Korea.

	Uzbekistan	South Korea	South-Central Asia	Sub-Saharan African Regions	East Asia	Western Europe	Northern America	Australia and New Zealand
Incidence	8.9	27.2	5.5	6.7–7.9	25.9	28.7	26.2	33.2
Mortality	5.2	7.8	3.2	5.1–5.4	11.8	10.2	8.2	9.5
Incidence per mortality	58.4%	28.7%	58.2%	68.4%–76.1%	45.6%	35.5%	31.3%	28.6%

All values correspond to the age-standardized rates per 100,000 population. Data source: The incidence and mortality rates are obtained from the 2020 GLOBOCAN.

## Data Availability

The data presented in this study are available in the article.

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
