# Peer review of "Challenges and Suggestions in the Management of Stomach and Colorectal Cancer in Uzbekistan: The Third Report of the Uzbekistan–Korea Oncology Consortium"

_ijerph, 2023, doi:10.3390/ijerph20085477_

Round 1

Reviewer 1 Report

Dear Authors, 

I am enthusiastic to review your manuscript, focused on a topic of interest to me, and providing research data that is in line with the current research in the field. I would like to take this chance to also congratulate you for your work and the effort put into the analysis of the management of stomach and colorectal cancer in Uzbekistan

In the review process, I would like to point out aspects that might bring a slight improvement to your manuscript in the following section-by-section assessment:

The Abstract serves as an overview of the topics addressed in the publication. Thereafter please do not mention the previous work of your research group in this section. There is no point in including the following phrases: 

“Therefore, the Ministry of Health and Welfare of Uzbekistan has formed an international advisory consortium that includes South Korean clinical oncologists and healthcare management experts. Our consortium envisioned a treatment and prevention strategy for six major cancer types with high mortality and morbidity rates in Uzbekistan. In two previous studies, we discussed the literature and data to identify effective treatment and prevention strategies for gynecological cancers (breast and cervical cancers) and cancers common in men (lung and liver cancers).”

Please completely remove such hints to other work 

Moreover, it would be beneficial to rewrite the section so that specific country related information that you have brought forwards with the review is highlighted.

Introduction:

For a review, the content of the introduction is not specific enough to the topic. It should be more focussed on colorectal and gastric cancer. Please reconsider what is included and rewrite accordingly! This introduction is too broad for the topic. 

Line 50: “the 70s and 80s” should be replaced with 8th and 9th decade of life or another rephrasing that clearly points out the age. 

Figure 1. Too low quality. Please replace with better resolution. Data source could be mentioned in the figure, before the caption, or within the caption, but not separately in a new paragraph. 

Gastric cancer

The section is well written, rather on point, concised and pointing out relevant aspects. However, please take care in expressing the risk factors, to mention that „drinking” refers to alcohol, not coffee/tea/carbonated drinks. Moreover, light vs strong alcohol are representing two different classes. 

Line 142: alcoholic beverages are measured glasses, generally not cups. 

Figure 2. Too low quality. Please replace with better resolution.

Colorectal cancer

The information is at times erred. Radiation therapy is recommended as PREOPERATIVE, not “perioperative”. Please correct the mistaken information. 

General summary and recommendations :

Please issue recommendations that are country specific and less broad than general recommendations of ESMO guidelines for these types of cancer. The work does not really stand out specifically covering these topics in Uzbekistan. 

Such recommendations could be formulated as: “Early screening is recommended starting with … years of age in males and … years in females, using FIT/FOBT with … frequency and colonoscopy… ” 

Please write down clear and concised recommendations that follow the generally accepted and implemented ones as required by the needs and possibilities of the healthcare system of the country in question

With these changes, if the work puts more focus on the region in question and skips presenting the other types of cancer which do not make the topic of the review, I do consider the work could be published. 

Best regards, 

Your Reviewer

Author Response

Reviewer #1

Dear Authors, 

I am enthusiastic to review your manuscript, focused on a topic of interest to me, and providing research data that is in line with the current research in the field. I would like to take this chance to also congratulate you for your work and the effort put into the analysis of the management of stomach and colorectal cancer in Uzbekistan

- We do appreciate your kind comment. We will do our best efforts to follow your considerate advices.

In the review process, I would like to point out aspects that might bring a slight improvement to your manuscript in the following section-by-section assessment:

  1. The Abstract serves as an overview of the topics addressed in the publication. Thereafter please do not mention the previous work of your research group in this section. There is no point in including the following phrases: 

“Therefore, the Ministry of Health and Welfare of Uzbekistan has formed an international advisory consortium that includes South Korean clinical oncologists and healthcare management experts. Our consortium envisioned a treatment and prevention strategy for six major cancer types with high mortality and morbidity rates in Uzbekistan. In two previous studies, we discussed the literature and data to identify effective treatment and prevention strategies for gynecological cancers (breast and cervical cancers) and cancers common in men (lung and liver cancers).”

Please completely remove such hints to other work. Moreover, it would be beneficial to rewrite the section so that specific country related information that you have brought forwards with the review is highlighted.

- Thank you for a valuable advice. We completely agree that it is not necessary to mention the previous study related, and the benefit to specific country related information brought forward. Therefore we removed the phrases you checked. Furthermore, we added the following sentence to enhance comprehensibility of the abstract; “Practical advice to increase the efficiency of treatment will be included, considering the current situation in Uzbekistan.” Line 34-36)

  1. Introduction:

For a review, the content of the introduction is not specific enough to the topic. It should be more focussed on colorectal and gastric cancer. Please reconsider what is included and rewrite accordingly. This introduction is too broad for the topic. 

- Thank you for your advice. Agreeing completely, we reduced the content regarding general view of cancer epidemiology (from two large paragraphs to a smaller single paragraph). Introductions to our consortium or references to past works have been deleted, because they are not necessarily shown at the front of the manuscript and it can be identified from the title, authors’ affiliation, and the subsequent descriptions. The second paragraph summarizes and introduces what to discuss in this review of gastric and colon cancer, and purpose of this review. The specific clinical contents of the diseases will be described in the following sections, so they are not covered in detail in the introduction. We hope that the revised introduction will follow the direction of the reviewer's comments. (Line 43-69 in revised manuscript)

2.1. Line 50: “the 70s and 80s” should be replaced with 8th and 9th decade of life or another rephrasing that clearly points out the age. 

- Thank you for your keen advice. We adjusted as you suggested. (70s and 80s => 8th and 9th decades of life)

2.2. Figure 1. Too low quality. Please replace with better resolution. Data source could be mentioned in the figure, before the caption, or within the caption, but not separately in a new paragraph. 

 - Thank you for your practical advice. Therefore we replaced the figure 1 with the full quality of which is adjustable. We add the reference in the caption (Data source: Data on the incidence and mortality are obtained from the 2020 GLOBOCAN [7-9].)

  1. Gastric cancer

The section is well written, rather on point, concised and pointing out relevant aspects. However, please take care in expressing the risk factors, to mention that „drinking” refers to alcohol, not coffee/tea/carbonated drinks. Moreover, light vs strong alcohol are representing two different classes. 

- Thank you for your keen advice. We clarified by adding ~alcohol after drinking at line 119 and 126 of revised manuscript.
It is an important point that considering the different classes of alcohol (e.g. beer, spirits, wine). There are studies that show that the incidence of gastric cancer varies depending on the type of alcohol, but the results are inconsistent. For example, in the study by Barstad et al., wine reduced the risk of GC, but in studies by Nomura, Zaride, and Everatt et al., the difference according to the type of alcohol was not clear. In the WCRF/AICR 3rd expert report, 'alcohol seems to cause cancer by ethanol regardless of type, and confounding factors may affect results that wine or light alcohol are less harmful. Most studies show that all types of alcoholic drinks increase risk of cancer.' Therefore, we additionally wrote in the manuscript, “Some studies reported that the increased risk of gastric cancer varies depending on the type of alcohol, but the results are inconsistent in the literature. The WCRF/AICR 3rd expert report stated that the risk of cancer increases with the amount of ethanol, regardless of the type of alcohol.’ Studies by Barstad, Nomura, Zaride, and Everatt were also added as references. (Line 128-132 in revised manuscript)

3.1. Line 142: alcoholic beverages are measured glasses, generally not cups. 

- Thank you for the keen comment. We adjusted as advised (cups => glasses, line 128 in revised manuscript)

3.2. Figure 2. Too low quality. Please replace with better resolution.

-  Agreeing your comment, we replaced figure 2 with a full quality one which is adjustable.

  1. Colorectal cancer

The information is at times erred. Radiation therapy is recommended as PREOPERATIVE, not “perioperative”. Please correct the mistaken information. 

- We appreciate your keen opinion. We adjusted it to preoperative.

  1. General summary and recommendations :

Please issue recommendations that are country specific and less broad than general recommendations of ESMO guidelines for these types of cancer. The work does not really stand out specifically covering these topics in Uzbekistan. 

Such recommendations could be formulated as: “Early screening is recommended starting with … years of age in males and … years in females, using FIT/FOBT with … frequency and colonoscopy… ” 

Please write down clear and concised recommendations that follow the generally accepted and implemented ones as required by the needs and possibilities of the healthcare system of the country in question

With these changes, if the work puts more focus on the region in question and skips presenting the other types of cancer which do not make the topic of the review, I do consider the work could be published. 

- Thank you for your valuable comments. We acknowledge that we have not clearly enough stated the purpose of this review. This article is the third article in a series of three articles. In the first review (Rim, et al. "Comparison of Breast Cancer and Cervical Cancer in Uzbekistan and Korea: The First Report of The Uzbekistan–Korea Oncology Consortium." Medicina 58.10 (2022): 1428.), the purpose of our project was described in detail. However, considering the opinions of reviewer 1 and another reviewer, we felt the need to clearly state the purpose of this review as well.

We added description of the target readers and purpose of this review from Line 66 to 70. In addition, the following contents have been added to the limitation. This content is in a similar vein with our response to the reviewer's query. (Line 407-416) “This review aims primarily to summarize comprehensive knowledge and advice on prioritizing health policy. However, this review may not be detailed enough for clinicians to refer to for actual treatment or diagnosis. Furthermore, we suggested the priority health needs in consideration of the socioeconomic situation of Uzbekistan. Therefore, topics related to the latest treatment or diagnosis that require a significant economic burden may not have been addressed enough. Although data are currently lacking, practice guidelines should be based on the experience of clinicians in Uzbekistan and domestic research data. Further details should be addressed in specific practice guidelines by professional groups including domestic clinicians.”

The contents of the screening period were originally thought to be too specific to be covered in this review. However, in agreement with the reviewer's opinion, a literature analysis on the screening period was added. (Line 191-200, Line 334-345) The setting of detailed recommendation of period seems to be the role of future practice guidelines produced by a specific expert group including domestic clinicians.

As the reviewer pointed out, we reduced or deleted the content irrelevant to the main topic (GC and CRC) of this review in the abstract and introduction. (Please refer to the answers to queries 1 and 2.)

We beg your understanding herein that we did our best effort to answer your query.

Reviewer 2 Report

1. Row 92 and row 95 - please add reference. Make abbreviation for gastric cancer - GC.

2. I consider that this review is not very clearly written. In my opinion, the authors focused too much on dietary habits and they neglected the precancerous lesions form upper and lower GI tract. 

3. They mention that in Uzbekistan there is lack of trained personal to perform endoscopy, so they should think of non-invasive biomarkers for detecting precancerous and cancerous formations. Please consider it!

4. Nowhere in the study was it mentioned at what age should screening endoscopy (gastro- and colonoscopy) started.

5. There is missing information of the type of endoscopy / whether they used chromoendoscopy/ as well as the biopsy protocols for screening.

6. It is mandatory to describe shortly a Correa cascade in HP section, as well as the type of test used to confirm the infection.

7. In the treatment section for GC- there is missing information abut HER receptor and consequent treatment.

Author Response

Reviewer #2

  1. Row 92 and row 95 - please add reference. Make abbreviation for gastric cancer - GC.

- Agreeing your comment, we added a reference at line 92 and 95.(line 82 an 84 in revised version). In addition, we found some minor error and corrected (fourth most common cancer type and fifth leading cancer-related death => fifth most common cancer type and fourth leading cancer-related death). We also used abbreviation GC for gastric cancer.

  1. I consider that this review is not very clearly written. In my opinion, the authors focused too much on dietary habits and they neglected the precancerous lesions form upper and lower GI tract. 

- We understand your point. We acknowledge that we have not clearly enough stated the purpose of this review. This article is the third article in a series of three articles. In the first review (Rim, et al. "Comparison of Breast Cancer and Cervical Cancer in Uzbekistan and Korea: The First Report of The Uzbekistan–Korea Oncology Consortium." Medicina 58.10 (2022): 1428.), the purpose of our project was described in detail. However, considering the opinions of reviewer 2 and another reviewer, we felt the need to clearly state the purpose of this review as well. Therefore, we have added description of target readers and purpose of this review from Line 66 to 70.

As described in Line 66-70, this review aims primarily to summarize comprehensive knowledge and advice on prioritizing health policy. In other words, this review is more of an advice resource for health policy than practice guidelines for clinicians. Therefore, this review focuses significantly on dietary habit and prevention. Uzbekistan has an annual GDP per capita of only USD 2000, and main food intake are red meat or processed red meat. Therefore, lifestyle modification and cancer prevention can be the effective measures to obtain health benefits.

This review may not be detailed enough for clinicians to refer to for actual treatment or diagnosis. Moreover, the target audience of this review includes not only doctors but also non-medical people including health policy makers. More details should be covered in practice guidelines produced by expert meetings including domestic clinicians. Of course, the clinical value of precancerous lesions is very important medical information. However, it seems to deviate somewhat from the primary purpose of this review. To help you understand, we have added a new limitation to the above (Lines 407-416).

We appreciate your expert opinion. You are welcomed to discuss with us about your opinion by email us. Or else, you may choose open reviewer so we can contact you to ask more advice in our future project.

  1. They mention that in Uzbekistan there is lack of trained personal to perform endoscopy, so they should think of non-invasive biomarkers for detecting precancerous and cancerous formations. Please consider it.

- We understand your point. Of course, non-invasive biomarker is an emerging field of cancer screening. We recognized that you are an expert in the diagnosis and treatment of digestive cancer. However, please consider that this review is a proposal to improve national health efficiency in consideration of Uzbekistan's social and economic situation. This review is based on our consortium's experience and the government report by IAEA & IARC & WHO investigating the medical situation in Uzbekistan. In Uzbekistan, endoscopy is a step where quality assurance for endoscope cleaning and disinfection is required. On the other hand, there are doctors who know how to handle endoscopy to some extent, even if they are not perfect experts from the perspective of international standards. We have to train them with a mid- to long-term plan. The field of pathology in Uzbekistan is very rudimentary. They primarily operate on classical microscopy-based pathological diagnosis. It will be difficult for them to efficiently perform biomarker-based screening in the near future. Uzbekistan's GDP per capita is $2,000. Therefore, we need to find a method that costs as little as possible and relies less on the medical supply of developed countries. We agree that your comment have a point. After conventional endoscope-based screening spreads to some extent, academic and medical discussions on biomarker screening can be begin in the future.

  1. Nowhere in the study was it mentioned at what age should screening endoscopy (gastro- and colonoscopy) started.

- Agreeing your comment, literature analyses on the screening period was added. (Line 191-200, Line 334-345)

  1. There is missing information of the type of endoscopy / whether they used chromoendoscopy/ as well as the biopsy protocols for screening.

- Thank you for pointing out a keen point. As mentioned in the newly added limitations section, this review is not a practice guideline, but rather a document advising on priorities for health policy decisions. This review aims to summarize and deal with comprehensive contents, and it is difficult to deal with details of procedures. Nonetheless, the reviewer's points are very important. Also, clinical practice and experience from Uzbekistan to date should be considered. Therefore, it should be dealt with in depth at an expert meeting including a gastroenterologist in Uzbekistan later. We will suggest to discuss about the detail of endoscopy in the future meeting.

  1. It is mandatory to describe shortly a Correa cascade in HP section, as well as the type of test used to confirm the infection.

- Agreeing your comment, we described about Correa cascade and related references at Line 179-182. Type of HP test is an important clinical issue. Nevertheless, the purpose of this review is to set health policy priorities and to provide comprehensive information to health care and non-health care practitioners. In addition, the practical aspect (economy, availability of products - In CIS countries, considerable procedures are required for importing foreign supplies) of Uzbekistan should be considered in discussing the type of test. It is recommended that the subject regarding type of test can be covered in future practice guidelines to be produced at an experts meeting including Uzbekistan gastroenterologists.

  1. In the treatment section for GC- there is missing information about HER receptor and consequent treatment.

- At Line 221-223, you can see we describe our opinion on the use of trastuzumab, citing the ToGa trial. The utilization of trastuzumab to obtain a significant survival benefit in HER2-positive gastric cancer patients is an important advance in gastric cancer treatment. However, the trastuzumab add-on costs thousands to tens of thousands of dollars. As we have stated in the purpose of this review, the purpose of this review is to set health policy priorities and take into account the social conditions of Uzbekistan. In Uzbekistan, conventional chemotherapy has only recently started to be offered to patients. Utilization of Trastuzumab will be difficult in the near future. Such resources should first be used for endoscopy examinations, prevention policies, and expansion of surgical facilities. This is because many Uzbek citizens could die without even getting a chance for basic screening or surgery. We added a reference to predict the economic burden of trastuzumab in Line 223. (Franchi M, Tritto R, Torroni L, Reno C, La Vecchia C, Corrao G. Effectiveness and Healthcare Cost of Adding Trastuzumab to Standard Chemotherapy for First-Line Treatment of Metastatic Gastric Cancer: A Population-Based Cohort Study. Cancers (Basel). 2020 Jun 25;12(6):1691)

Reviewer 3 Report

Chai Hong et al., in this manuscript “Challenges and suggestions in the management of stomach and colorectal cancer in Uzbekistan: The third report of the Uzbekistan-Korea oncology consortium” summarized the management strategies for gastric and colorectal cancers in Uzbekistan. In particular, gastrointestinal cancers can be significantly prevented by certain screening strategies. I personally enjoy reading it and hope will attract attention from other researchers involve in gastrointestinal cancers. Following some comments to improve the quality of this review.

Major comment:

No

Minor comments:

1)      Improve quality of figure 1.

2)      Improve quality of figure 2.

3)      Line 413 – 414 : “[“ ?

4)      Delete “[Internet]” at lines 422 and 426

Author Response

Reviewer #3

Chai Hong et al., in this manuscript “Challenges and suggestions in the management of stomach and colorectal cancer in Uzbekistan: The third report of the Uzbekistan-Korea oncology consortium” summarized the management strategies for gastric and colorectal cancers in Uzbekistan. In particular, gastrointestinal cancers can be significantly prevented by certain screening strategies. I personally enjoy reading it and hope will attract attention from other researchers involve in gastrointestinal cancers. Following some comments to improve the quality of this review.

Authors reply: We do appreciate your thorough review and considerate opinions. We adjusted following your queries.

Major comment:

No

Minor comments:

1)      Improve quality of figure 1.

- We provided full quality (original chart file) figure 1 in the word file.

2)      Improve quality of figure 2.

- We provided full quality (original chart file) figure 2 in the word file.

3)      Line 413 – 414 : “[“ ?

- Thank you for your keen consideration. We checked mistyped “[“, and the reference itself is removed during revision to make introduction brief.

4)      Delete “[Internet]” at lines 422 and 426

- Thank you for your through review. We removed [internet].

Round 2

Reviewer 1 Report

The manuscript has improved significantly and I would now like to recommend it for publication.

Reviewer 2 Report

Nessecary revisions gave been done